# Exome sequencing of familial high-grade serous ovarian carcinoma reveals heterogeneity for rare candidate susceptibility genes

Deepak N. Subramanian [1,2], Magnus Zethoven[3], Simone McInerny[4], James A. Morgan [4],
Simone M. Rowley[1], Jue Er Amanda Lee [1,2], Na Li [1,2], Kylie L. Gorringe [2], Paul A. James [1,2,4,5] &
Ian G. Campbell [1,2,5 ✉]

High-grade serous ovarian carcinoma (HGSOC) has a significant hereditary component, approximately half of which cannot be explained by known genes. To discover genes, we analyse germline exome sequencing data from 516 *BRCA1/2*-negative women with HGSOC, focusing on genes enriched with rare, protein-coding loss-of-function (LoF) variants. Overall, there is a significant enrichment of rare protein-coding LoF variants in the cases ($p < 0.0001$, chi-squared test). Only thirty-four (6.6%) have a pathogenic variant in a known or proposed predisposition gene. Few genes have LoF mutations in more than four individuals and the majority are detected in one individual only. Forty-three highly-ranked genes are identified with three or more LoF variants that are enriched by three-fold or more compared to GnomAD. These genes represent diverse functional pathways with relatively few involved in DNA repair, suggesting that much of the remaining heritability is explained by previously under-explored genes and pathways.

[1] Cancer Genetics Laboratory, Peter MacCallum Cancer Centre, Melbourne, VIC 3000, Australia. [2] Sir Peter MacCallum Department of Oncology, The University of Melbourne, Parkville, VIC 3010, Australia. [3] Bioinformatics Core Facility, Peter MacCallum Cancer Centre, Melbourne, VIC 3000, Australia. [4] The Parkville Familial Cancer Centre, Peter MacCallum Cancer Centre and The Royal Melbourne Hospital, Melbourne, VIC 3000, Australia. [5] These authors contributed equally: Paul A. James, Ian G. Campbell. ✉email: ian.campbell@petermac.org

Epithelial ovarian carcinoma is a heterogeneous disease, representing approximately 3.7% of all new female cancer diagnoses[1]. It comprises several distinct histological sub-types (including high- and low-grade serous, clear cell, endometrioid and mucinous), each one displaying different behaviours at both the clinical and molecular levels[2]. Around 70% of epithelial ovarian tumours are high-grade serous ovarian carcinomas (HGSOC), which are relatively aggressive and have a poor prognosis.

There is a significant genetic component to the risk of ovarian carcinoma[3], with germline mutations in *BRCA1* and *BRCA2* identifiable in 11−23% of affected women with HGSOC[4,5], rising to as high as 42% of affected women with a family history of two or more ovarian carcinomas[6]. Other genes make a smaller contribution to HGSOC risk (e.g. *RAD51C, RAD51D, BRIP1*[7−11]), but the hereditary basis of approximately 50% of cases remains unexplained[3], which compromises risk management for these women and their families.

Efforts to identify additional moderate-to-high-risk hereditary breast and ovarian cancer (HBOC) genes have largely been restricted to candidate gene approaches using targeted next-generation sequencing (NGS) panels of known cancer predisposition genes[12−18], which have collectively only resolved a very small proportion of unexplained families. Although three studies utilised data from whole-exome sequencing (WES) of *BRCA1* and *BRCA2*-negative ovarian carcinoma patients[19−21], these analysed only a subset of candidate genes in the available data and included non-HGSOC tumour types in their case cohorts. Others utilised germline sequencing data from The Cancer Genome Atlas (TCGA)[22−25], but this approach is limited by the diverse

technologies used to generate TCGA data along with the absence of any linked family history information. None of the previous studies have identified candidate HBOC genes that have been validated in multiple independent studies; nor has there been any consistency of the candidates identified across different studies.

As a first step in resolving the missing heritability of ovarian carcinoma, we present WES data from a large cohort of women diagnosed with HGSOC, who were tested through a familial cancer clinic but returned negative findings for the *BRCA1* and *BRCA2* genes. Our results indicate that familial HGSOC is enriched for rare protein-coding loss-of-function (LoF) variants, but displays high genetic heterogeneity, with no single proposed candidate gene identified in our cohort found in more than 2.4% of cases. These genes are functionally diverse, with only a small number associated with DNA repair as with other known HGSOC predisposition genes, suggesting that much of the remaining missing heritability may lie in genes and pathways that are currently overlooked.

## Results

**Exome sequencing and variant filtering.** Whole-exome sequencing was successfully performed on all germline DNA samples to an average depth of 126× with 98.4% of the bases covered to >20×. Principal component analysis (PCA) was performed using common single nucleotide polyporphisms (SNPs), demonstrating that over 95% of participants were of Western European origin (Supplementary Fig. 1). Numerous quality and variant frequency filters (as summarised in Fig. 1) were applied to the data to remove artefacts, common variants and lower-impact

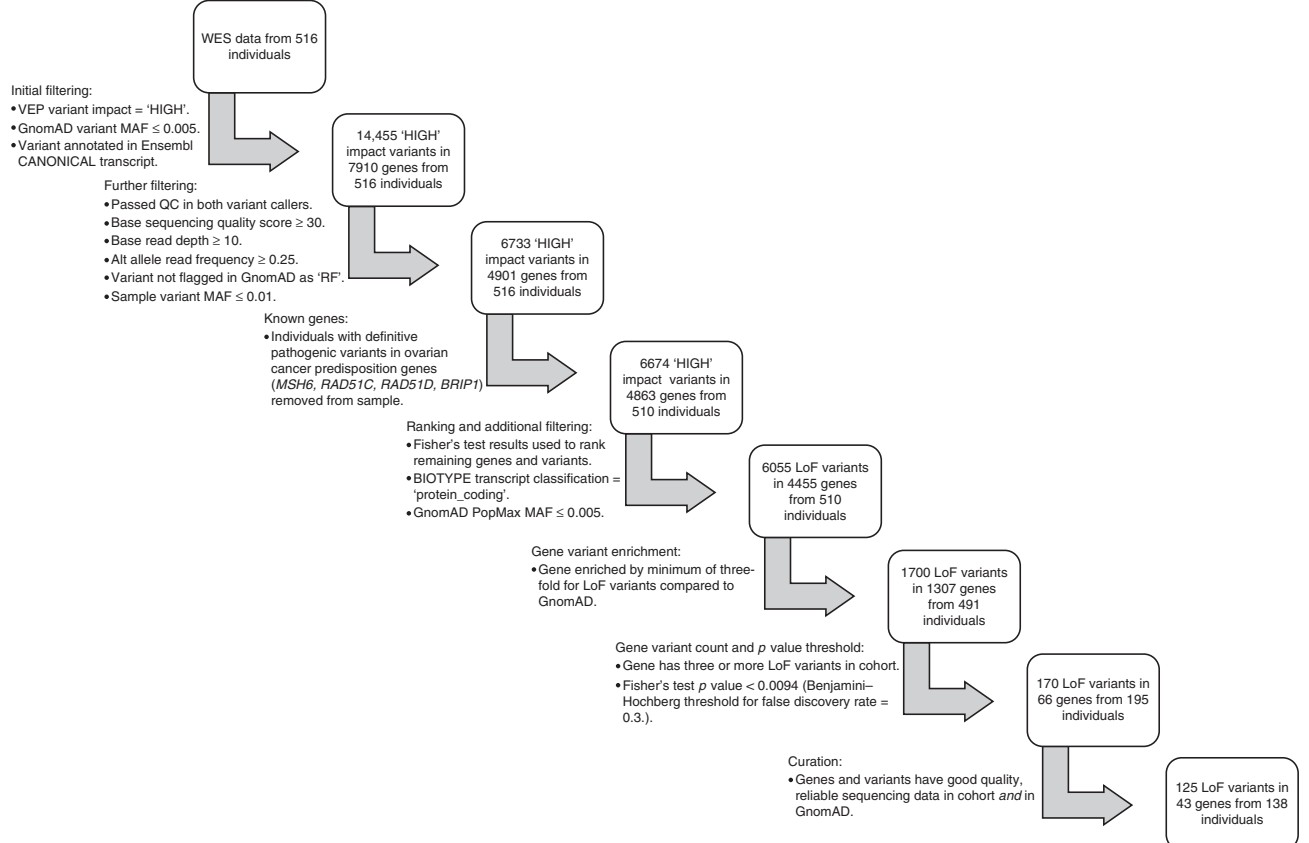

**Fig. 1 Flowchart illustrating the filtering, ranking, prioritisation and curation steps used on the processed exome variant (vcf) data.** Steps performed in the post-sequencing pipeline (i.e. alignment of FASTQ reads, variant calling and annotation) are not displayed. Numerical figures refer to unique variants and genes. *LoF* loss of function, *VEP* Variant Effect Predictor, *MAF* minor allele frequency, *RF* failed random forests filter.

**Table 1 Known ovarian carcinoma predisposition genes with loss-of-function (LoF) and deleterious missense variants in the total case cohort.**

| Gene | No. of cases (%) | Variants in cases | | |
|------|------------------|-------------------|--|--|
| | | Transcript | Protein | Consequence |
| MSH6[a] | 1 (0.19) | c.2731 C > T | p.Arg911Ter | Stop-gain |
| RAD51C | 3 (0.58) | c.313delT | p.Ser105GlnfsTer3 | Frameshift |
| | | c.145 + 1_145 + 2insC | — | Splice donor |
| | | c.773 G > A | p.Arg258His | Missense |
| RAD51D | 1 (0.19) | c.556 C > T | p.Arg186Ter | Stop-gain |
| BRIP1 | 1 (0.19) | c.1372 G > T | p.Glu458Ter | Stop-gain |
| Total | 6 (1.2) | | | |

[a]Other Lynch syndrome genes (MLH1, MSH2, PMS2) had no deleterious variants present within the cohort.

**Table 2 Proposed ovarian carcinoma predisposition genes with loss-of-function (LoF) and known deleterious missense variants in the discovery case cohort.**

| Gene | No. of pathogenic alleles in cases[a] (%) | No. of LoF alleles in GnomAD[b] (%) | OR (95% CI)[c] | P value[c] |
|------|------------------------------------------|-------------------------------------|----------------|------------|
| ATM | 7 (0.69)[d] | 195 (0.17) | 2.98 (0.95−7.1) | 0.030 |
| BLM | 3 (0.29)[e] | 131 (0.11) | 2.66 (0.54−7.96) | 0.11 |
| CHEK2 | 2 (0.20) | 401 (0.34) | 0.57 (0.07−2.09) | 0.59 |
| FANCM | 4 (0.39) | 344 (0.29) | 1.35 (0.36−3.49) | 0.55 |
| MRE11A | 3 (0.29) | 57 (0.048) | 6.11 (1.22−18.8) | 0.015 |
| NBN | 1 (0.098) | 90 (0.076) | 1.29 (0.03−7.37) | 0.54 |
| NF1 | 1 (0.098) | 19 (0.016) | 6.10 (0.15−38.5) | 0.16 |
| PALB2 | 3 (0.29) | 86 (0.073) | 4.05 (0.82−12.3) | 0.041 |
| RAD50 | 2 (0.20) | 168 (0.14) | 1.38 (0.17−5.08) | 0.66 |
| RECQL | 2 (0.20) | 289 (0.24) | 0.80 (0.10−2.92) | 1.0 |
| Total | 28 (2.7) | | | |

[a]Total $n = 1020$ alleles tested per gene.
[b]GnomAD non-Finnish European (NFE), non-cancer sub-population.
[c]Fisher's exact test results (OR odds ratio, CI confidence interval). Calculations exclude missense variants.
[d]Figure for ATM includes two missense variants (c.7271 T > G, c.8147 T > C) classed as 'pathogenic' in NCBI ClinVar.
[e]Figure for BLM excludes additional stop-gain variant (c.2208 T > G) found in cis with frameshift variant (c.2206dupT) in the same individual.

variants that are unlikely to represent moderate-to-high-risk alleles. Implementing these filters left 6733 unique, rare 'HIGH' impact variants in 4901 genes.

**Variants in known and proposed ovarian carcinoma risk genes.** Sequence data were analysed for deleterious variants in known ovarian carcinoma predisposition genes, including RAD51C, RAD51D[7,8], BRIP1[9] and the Lynch syndrome genes (MLH1, MSH2, MSH6, PMS2)[5]. As expected, no BRCA1 or BRCA2 variants were identified in this pre-screened group and only six of the 516 cases (1.2%) had clinically actionable variants in one of the other genes (Table 1). Five individuals carried LoF variants in one of MSH6, RAD51C, RAD51D or BRIP1, and one had a likely pathogenic missense variant in RAD51C[26,27]. These six cases were removed from the discovery cohort, since the presence of deleterious variants in one of these genes is likely to explain their personal and family history of cancer.

Amongst the remaining 510 cases, 28 individuals (5.5%) had a LoF or known deleterious missense variant in 16 genes that have been proposed as ovarian cancer predisposition genes and are commonly included on HBOC gene testing panels (Table 2). After applying Fisher's exact tests as described below, only PALB2, ATM and MRE11A were enriched for LoF variants in the cases compared to GnomAD, although the number of variants and cases was small, and caution should be exercised interpreting the odds ratios as risk estimates. As it is currently unclear whether variants in these genes have a genuine role in HGSOC

predisposition, these individuals were retained in the discovery cohort for subsequent analysis.

**Analysis of ranked candidate genes and variants of interest.** To assess for variant enrichment, the gene-level frequency of 'HIGH' impact variants in the remaining 510 cases was compared to the gene-level frequency in the GnomAD sub-population ($n = 59,095$), as detailed in the Methods. Overall, for all protein-coding genes represented on the WES panel ($n = 19,818$), there was a significantly higher number of rare LoF variants in the cases compared to GnomAD ($p < 0.0001$, chi-squared test). Two-tailed Fisher's exact tests were performed to rank genes by level of enrichment (as represented by their $p$ values), and plotting their distribution (Fig. 2) demonstrated a significantly greater number of genes enriched for rare LoF variants ($n = 133$, OR > 1 and $p < 0.01$) compared to genes depleted for rare LoF variants ($n = 19$, OR < 1 and $p < 0.01$) in the cases vs. GnomAD ($p < 0.0001$, chi-squared test).

To identify the most likely candidates with an excess of LoF variants from amongst the remaining 4863 genes (Supplementary Data 1), a number of additional steps were applied (Fig. 1). First, the Benjamini−Hochberg procedure[28] for multiple testing was used on the ranked list of Fisher's test $p$ values to establish a 'discovery' threshold of 0.0094 (number of $p$ values = 4863, false discovery rate = 0.3). Next, only protein-coding genes enriched with rare LoF variants (in any of the major GnomAD sub-populations) by at least three-fold in the cases were retained,

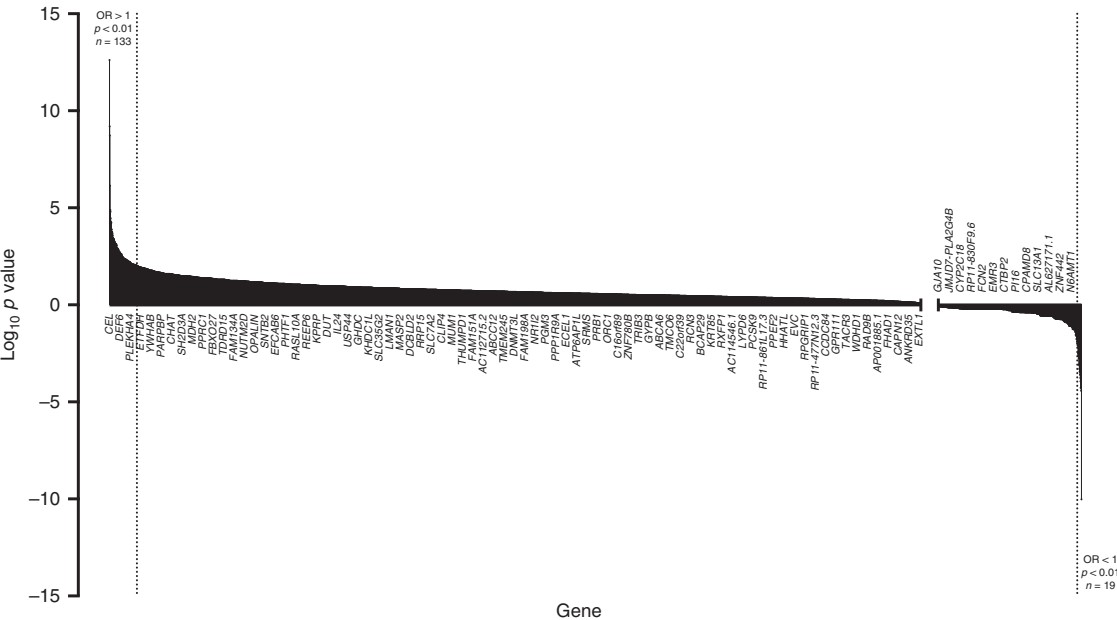

**Fig. 2 Waterfall bar chart displaying the degree of enrichment for all protein-coding genes represented on the WES panel ($n = 19{,}818$) with filtered LoF variants in comparison to GnomAD, ordered by decreasing $\log_{10} p$ value. Any negative $\log_{10} p$ values for genes with calculated odds ratios >1 were transformed into positive values prior to plotting.** Genes with $\log_{10} p$ values of 0 (equivalent to a $p$ value of 1 i.e. no difference to GnomAD frequency) were not plotted. Shaded areas to left and right of dashed lines represent genes with $p$ values < 0.01 and odds ratios >1 ($n = 133$) or <1 ($n = 19$), respectively. Genes are labelled on the $x$-axis every 50 rows from the ordered list for illustration only.

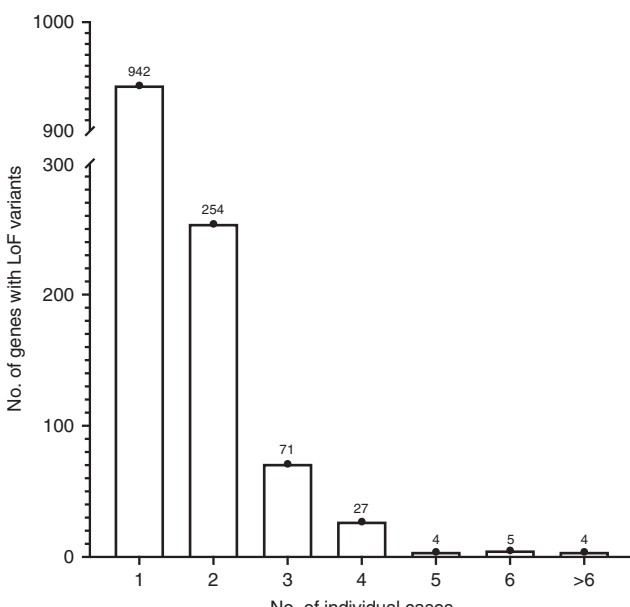

**Fig. 3 Frequency bar chart displaying number of genes containing LoF variants vs. the number of individual cases they were present in.** Total number of genes stated above bar for each class of number of cases.

reducing the list to 1700 unique LoF variants in 1307 genes amongst 491 individuals. Of these genes, the vast majority had a LoF variant in just one individual ($n = 942$) with most of the remainder occurring in 2−4 individuals (Fig. 3). Finally, genes with LoF variants in three or more individuals and $p$ values below the calculated multiple testing threshold ($n = 66$) were prioritised for curation, including detailed GnomAD and bam file review. Twenty-three genes with low-confidence LoF variants were removed during curation; these included 15 genes that were removed due to their remaining valid variants occurring in fewer than three individuals, or falling below our three-fold enrichment threshold.

The remaining 43 highest-ranked candidate genes with high-confidence, rare LoF variants are displayed in Table 3 (for individual variants and associated case data, refer to Supplementary Data 2). The top-ranked genes are involved in very diverse functional pathways (e.g. transporter proteins and metabolic enzymes), and of note, few appear to have a role in DNA repair despite the fact that all known HBOC genes to date are directly or indirectly involved with that function[2,29]. The majority of these candidate genes have not been reported to contain pathogenic somatic mutations in serous ovarian tumour samples from the COSMIC database (Table 3), and for those that do, the frequency of somatic variants is low (<1% of samples). Comparing the family history distribution of candidate gene carriers ($n = 138$) and non-carriers ($n = 378$), there was no significant difference in the likelihood of being a carrier in those with a family history of breast and/or ovarian cancer in one or more first- or second-degree relatives vs. those with no family history ($p = 0.55$, Fisher's exact test).

To assess if this reflected a genuine lack of enrichment of DNA repair genes, the total frequency of rare LoF variants in DNA repair genes grouped by functional pathway[30] amongst the cases in the discovery cohort was compared with the GnomAD sub-population (Table 4), excluding known HBOC genes that were previously searched for in the total case cohort (Table 1). One-hundred-and-five cases (21%) harboured at least one LoF variant across all DNA repair and associated genes, but the total frequency of LoF variants across all functional categories in the cases was very similar to GnomAD (0.063% vs. 0.061%, $p = 0.60$, Fisher's exact test). Although the frequency of LoF variants in the subset of genes involved in the nucleotide excision repair, homologous recombination repair, Fanconi anaemia and non-homologous end-joining pathways were higher in the cases vs. GnomAD, only the homologous recombination repair category was significantly enriched ($p = 0.032$, Fisher's exact test).

**Table 3 Top-ranked candidate genes remaining after curation with loss-of-function (LoF) variants present in three or more discovery case cohort individuals.**

| Gene | Protein function[a] | No. of LoF alleles in cases[b] (%) | No. of LoF alleles in GnomAD[c] (%) | OR (95% CI)[d] | p value[d] | No. of COSMIC cases with pathogenic somatic tumour mutations[e] (%) |
|---|---|---|---|---|---|---|
| SLC12A4[134] | K/Cl transporter | 6 (0.59) | 54 (0.046) | 12.9 (4.54–30.1) | 0.000013 | 0 (0) |
| MAP6D1 | Microtubule stabilisation | 3 (0.29) | 9 (0.0076) | 38.6 (6.72–156) | 0.00013 | 0 (0) |
| SORD | Sorbitol metabolism | 6 (0.59) | 89 (0.075) | 7.85 (2.80–17.8) | 0.00018 | 0 (0) |
| CCDC88B | Apoptosis inhibition | 5 (0.49) | 57 (0.048) | 10.2 (3.19–25.3) | 0.00020 | 1 (0.18) |
| CPT1B | Mitochondrial metabolic transport | 4 (0.39)[g] | 36 (0.031) | 12.9 (3.33–36.2) | 0.00038 | 1 (0.18) |
| ZBTB45 | Unknown | 4 (0.39) | 37 (0.031) | 12.6 (3.25–35.0) | 0.00042 | 0 (0) |
| LOXL2 | Elastin fibre remodelling | 4 (0.39) | 39 (0.033) | 11.9 (3.09–33.2) | 0.00050 | 0 (0) |
| SSX3 | Gonadal transcription repressor | 3 (0.29) | 11 (0.013) | 22.6 (4.04–85.8) | 0.00056 | 0 (0) |
| ZCCHC4 | Ribosomal RNA methylation | 12 (1.2) | 444 (0.38) | 3.14 (1.61–5.57) | 0.00070 | 2 (0.36) |
| RPA3[131] | DNA damage checkpoint signalling | 3 (0.29) | 18 (0.015) | 19.4 (3.65–66.5) | 0.00074 | 0 (0) |
| MMAA | Mitochondrial cobalamin transport | 5 (0.49) | 78 (0.066) | 7.46 (2.35–18.2) | 0.00076 | 1 (0.18) |
| IMPDH2[135-37] | Guanine nucleotide synthesis | 3 (0.29) | 19 (0.016) | 18.3 (3.47–62.3) | 0.00085 | 0 (0) |
| GPALPP1 | Unknown | 3 (0.29) | 19 (0.016) | 18.3 (3.47–62.3) | 0.00086 | 0 (0) |
| WRAP53[173-75] | Telomere maintenance/p53 regulation | 4 (0.39) | 49 (0.041) | 9.49 (2.48–26.0) | 0.0011 | 0 (0) |
| STARD6 | Unknown | 3 (0.29) | 22 (0.019) | 15.8 (3.03–52.8) | 0.0013 | 1 (0.18) |
| LLGL2 | aPKC regulation | 4 (0.39) | 55 (0.047) | 8.45 (2.22–23.0) | 0.0017 | 1 (0.18) |
| FBLIM1 | Cell-extracellular matrix regulator | 3 (0.29) | 26 (0.022) | 13.4 (2.59–43.8) | 0.0019 | 0 (0) |
| CDH23 | Intercellular adhesion | 3 (0.29) | 26 (0.022) | 13.3 (2.58–43.7) | 0.0020 | 2 (0.36) |
| IFIT2[76] | Apoptosis stimulator | 4 (0.39) | 58 (0.049) | 7.99 (2.10–21.6) | 0.0020 | 1 (0.18) |
| FAM216A | Unknown | 4 (0.39) | 60 (0.051) | 7.75 (2.04–21.0) | 0.0023 | 0 (0) |
| MIPOL1[77] | Developmental role | 4 (0.39) | 60 (0.051) | 7.74 (2.04–20.9) | 0.0023 | 1 (0.18) |
| LTBP1 | TGF-beta targeting | 3 (0.29) | 28 (0.024) | 12.4 (2.42–40.4) | 0.0023 | 4 (0.71) |
| TBXAS1 | Thromboxane A2 synthesis | 4 (0.39) | 61 (0.052) | 7.62 (2.01–20.6) | 0.0024 | 2 (0.36) |
| CCDC14 | Centriole regulator | 6 (0.59) | 150 (0.13) | 4.65 (1.67–10.4) | 0.0024 | 0 (0) |
| TTC24 | Unknown | 3 (0.29) | 32 (0.027) | 10.8 (2.12–34.7) | 0.0034 | 0 (0) |
| DLGAP5[78,79] | Mitotic microtubule protein | 3 (0.29) | 33 (0.028) | 10.6 (2.07–33.8) | 0.0036 | 1 (0.18) |
| SLC38A8 | Glutamine-H+ antiporter | 3 (0.29) | 34 (0.029) | 10.2 (2.01–32.7) | 0.0039 | 0 (0) |
| CARML2 | Intracellular signalling | 3 (0.29) | 36 (0.031) | 9.63 (1.89–30.5) | 0.0046 | 1 (0.18) |
| ANKAR | Unknown | 6 (0.59) | 174 (0.15) | 4.01 (1.45–8.94) | 0.0048 | 0 (0) |
| SCYL3 | Cell adhesion and migration | 3 (0.29) | 37 (0.031) | 9.42 (1.85–29.8) | 0.0049 | 0 (0) |
| ZNF418 | MAPK signalling repressor | 5 (0.49) | 127 (0.11) | 4.58 (1.46–11.0) | 0.0058 | 0 (0) |
| USP50[132] | Cell cycle arrest after DNA damage | 4 (0.39) | 80 (0.068) | 5.78 (1.53–15.4) | 0.0061 | 0 (0) |
| RAD1[f,33] | DNA damage-sensing cell cycle inhibitor | 5 (0.49) | 129 (0.11) | 4.51 (1.44–10.8) | 0.0061 | 0 (0) |
| RASSF7[f,80-82] | Mitotic microtubule regulator | 3 (0.29) | 41 (0.035) | 8.49 (1.68–26.7) | 0.0064 | 0 (0) |
| PLEKHA4 | Unknown | 4 (0.39) | 82 (0.069) | 5.67 (1.51–15.2) | 0.0065 | 0 (0) |
| LRRC56 | Unknown | 3 (0.29) | 44 (0.037) | 7.91 (1.57–24.8) | 0.0077 | 0 (0) |
| HARS2 | Mitochondrial tRNA synthesis | 3 (0.29) | 45 (0.038) | 7.74 (1.54–24.2) | 0.0081 | 3 (0.53) |
| ADGRD1 | Unknown | 3 (0.29) | 46 (0.039) | 7.58 (1.50–23.6) | 0.0086 | 1 (0.18) |
| PRKACG | cAMP-dependent kinase subunit | 3 (0.29) | 46 (0.039) | 7.57 (1.50–23.6) | 0.0086 | 0 (0) |
| CDKL3[f,83] | Cyclin-dependent kinase-like protein | 4 (0.39) | 90 (0.077) | 5.13 (1.37–13.7) | 0.0090 | 0 (0) |
| VSIG1[f,84] | Junctional adhesion molecule | 3 (0.29) | 33 (0.039) | 7.54 (1.48–24.1) | 0.0091 | 1 (0.18) |
| ZNF616 | Unknown | 4 (0.39) | 91 (0.077) | 5.11 (1.36–13.6) | 0.0092 | 0 (0) |
| ANKRD18A | Unknown | 3 (0.29) | 47 (0.040) | 7.38 (1.46–23.0) | 0.0092 | 0 (0) |

[a]Protein function information sourced from NCBI Gene, OMIM and references contained therein.
[b]Total n = 1020 alleles tested per gene.
[c]GnomAD non-Finnish European (NFE), non-cancer sub-population.
[d]Fisher's exact test results (OR odds ratio, CI confidence interval). Genes ranked by ascending p value.
[e]Confirmed somatic serous ovarian tumour sample mutation data for each candidate gene (sourced from COSMIC v90[38], n = 562 samples) includes pathogenic missense, splice site, frameshift and stop-gain mutations.
[f]Gene with data (as referenced) supporting a role in tumourigenesis.
[g]Two variants found in one individual—phase unknown.

**Table 4 List of DNA repair genes with loss-of-function (LoF) variants in the discovery case cohort, grouped by pathway.**

| Repair pathway | Genes[a] | No. of LoF alleles in cases[b] (%) | No. of LoF alleles in GnomAD[c] (%) | OR (95% CI)[d] | p value[d] |
|---|---|---|---|---|---|
| Base excision repair | RFC1, RFC2, RFC3, RFC4, RFC5, XRCC1, PCNA, PARP1, PARP2, APEX1, FEN1, POLE, POLD1, LIG3, OGG1, UNG, SMUG1, MBD4, TDG, MUTYH, NTHL1, MPG, NEIL1, NEIL2, NEIL3, APEX2, PNKP, APLF, PARP3, LIG1 | 16 (0.052) | 2146 (0.061) | 0.854 (0.49–1.39) | 0.64 |
| Mismatch repair | RFC1, RFC2, RFC3, RFC4, RFC5, PCNA, MSH3, MSH4, MSH5, MLH3, PMS1, PMS2P3, EXO1, POLD1 | 4 (0.028) | 1041 (0.063) | 0.439 (0.12–1.13) | 0.095 |
| Nucleotide excision repair | RPA1, RPA2, RPA3, RPA4, RFC1, RFC2, RFC3, RFC4, RFC5, PCNA, ERCC1, ERCC4, ERCC2, ERCC5, XPC, ERCC6, GTF2H2, ERCC3, XPA, RAD23B, POLE, POLD1, RAD23A, LIG3, CETN2, DDB1, DDB2, GTF2H1, GTF2H3, GTF2H4, GTF2H5, CDK7, CCNH, MNAT1, LIG1, ERCC8, UVSSA, XAB2, MMS19 | 17 (0.043) | 1691 (0.037) | 1.15 (0.67–1.84) | 0.51 |
| Homologous recombination repair | ATM, RPA1, RPA2, RPA3, RPA4, RAD51, NBN, RAD50, CHEK2, PALB2, MUS81, EME1, MRE11A, RAD52, RAD51B, DMC1, XRCC2, XRCC3, RAD54L, RAD54B, SHFM1, RBBP8, SLX1A, SLX1B, GEN1 | 23 (0.090) | 1614 (0.056) | 1.62 (1.02–2.43) | 0.032 |
| Fanconi anaemia genes | ATR, ERCC1, FANCA, FANCB, FANCC, FANCD2, FANCE, FANCF, FANCG, FANCI, FANCL, FANCM, PALB2, CHEK1, SLX4, FAN1, FAAP20, FAAP24, MUS81, EME1 | 24 (0.12) | 1980 (0.090) | 1.39 (0.89–2.07) | 0.12 |
| Non-homologous end-joining | XRCC6, XRCC5, PRKDC, LIG4, XRCC4, DCLRE1C, NHEJ1 | 3 (0.042) | 275 (0.033) | 1.27 (0.26–3.74) | 0.52 |
| Direct DNA damage reversal | MGMT, ALKBH2, ALKBH3 | 1 (0.033) | 378 (0.11) | 0.305 (0.01–1.71) | 0.39 |
| All unique DNA repair genes | | 77 (0.066) | 8268 (0.063) | 1.06 (0.84–1.33) | 0.59 |
| Other associated genes, i.e. indirect regulators of genomic stability | PAXIP1, BLM[e], MLL3, CRIP1, CDK12, BAP1, BARD1, WRN, BUB1, CENPE, ZW10, TTK, KNTC1, AURKB, POLB, POLH, POLQ, TDP1, TDP2, NUDT1, DUT, RRM2B, POLG, REV3L, MAD2L2, REV1, POLI, POLK, POLL, POLM, POLN, TREX1, TREX2, APTX, SPO11, ENDOV, UBE2A, UBE2B, RAD18, SHPRH, HLTF, RNF168, SPRTN, RNF8, RNF4, UBE2V2, UBE2N, H2AFX, CHAF1A, SETMAR, RECQL4, MPLKIP, DCLRE1A, DCLRE1B, PRPF19, RECQL, RECQL5, HELQ, RDM1, NABP2, ATRIP, MDC1, RAD1, RAD9A, HUS1, RAD17, TP53, TP53BP1, TOPBP1, CLK2, PER1 | 42 (0.057) | 4733 (0.056) | 1.02 (0.73–1.38) | 0.88 |
| Total (all DNA repair and other associated genes) | | 119 (0.063) | 13,001 (0.061) | 1.05 (0.87 to 1.25) | 0.60 |

[a]Genes and their associated pathways sourced from list curated by Chae et al.[30]. Table and calculations exclude known HBOC genes (BRCA1, BRCA2, MLH1, MSH2, MSH6, PMS2, RAD51C, RAD51D, BRIP1), as samples with pathogenic LoF variants in these genes were removed from the cohort during filtering.
[b]Total n = 1020 multiplied by number of genes in pathway.
[c]GnomAD non-Finnish European (NFE), non-cancer sub-population.
[d]Fisher's exact test results (OR odds ratio, CI confidence interval).
[e]Figure for BLM excludes additional stop-gain variant (c.2208 T > G) found in cis with frameshift variant (c.2206dupT) in the same individual.

## Discussion

Reported here is the largest WES study to date of HGSOC patients with no detectable *BRCA1* or *BRCA2* germline mutations. The extreme degree of genetic heterogeneity underlying HGSOC predisposition is demonstrated by the fact that 1307 genes are enriched for LoF variants by a minimum of three-fold, along with the fact that amongst the 43 high-priority candidates, the median number of LoF variants was only four. Although a proportion of these genes are likely to be false positives, the fact there is a significantly higher number of rare LoF variants in the case cohort compared to GnomAD as well as significantly more genes with ORs > 1 compared to those with ORs < 1 indicates that the list likely includes many genuine HGSOC predisposition genes.

Among the top-ranked genes (Table 3), a small number function in a manner analogous to other known tumour suppressor genes. For example, *RPA3*[31], *USP50*[32] and *RAD1*[33] are thought to participate in arresting cell cycle progression in response to DNA damage. Others, such as *SLC12A4* (a potassium and chloride ion co-transporter)[34] and *IMPDH2* (the rate-limiting enzyme in guanine nucleotide synthesis)[35–37], are known to have an oncogenic role in various tumour types. Assuming their biological function as described in the literature is accurate and complete, it is unclear how germline LoF variants in these genes might predispose to tumour development. However, the vast majority of top-ranked genes either have no known role in tumorigenesis (e.g. *LOXL2*) and/or their function is currently unknown (e.g. *ZBTB45*). This uncertainty suggests that approaches to gene discovery that emphasise candidate gene function above other considerations (such as relative frequency of LoF mutations in cases vs. controls) may fail to identify HGSOC predisposition genes functioning in pathways other than those classically inactivated in HBOC, such as DNA repair pathway genes. Of these, only homologous recombination repair pathway genes were modestly enriched for rare LoF variants in the cohort, indicating that mutations in these genes cannot alone explain the missing heritability of HGSOC.

Only 16 top-ranked genes had any somatic mutations recorded in COSMIC (Table 3), none of which exceeded 1% of serous ovarian tumour samples present in the database. This is consistent with the finding that established germline susceptibility genes, with the exception of *TP53*, are also rarely found to harbour somatically-acquired mutations in sequenced tumour samples. *BRCA1* pathogenic somatic mutations, for example, are only present in 1.59% of serous ovarian carcinoma samples in the COSMIC database[38].

Only a small fraction of cases (6.6%) were potentially explained by genes known or suggested to be linked with a higher risk of HGSOC, which is consistent with the low frequency reported in other studies[5,10,11,13,14,17,39]. Of note, there was no enrichment for Lynch syndrome genes (*MLH1*, *MSH2*, *MSH6*, *PMS2*[40]), despite the large size of the cohort. This reflects the fact that the ovarian tumour types most often associated with Lynch syndrome (i.e. clear cell and low-grade endometrioid)[41] were not represented in this patient group.

Although many of the suspected HBOC-associated genes harboured LoF variants (Table 2), the frequencies were low and only *PALB2*, *ATM* and *MRE11A* showed some degree of enrichment compared to GnomAD. The level of enrichment was relatively modest, with very wide confidence intervals due to the small numbers present, making it challenging to interpret their true significance. Previous work suggested a similarly modest increase in risk for *ATM* and *PALB2*, but not for *MRE11A*[13,14], casting doubt on whether the latter is truly an ovarian carcinoma predisposition gene. Recent additional data from *PALB2* families found that pathogenic variants are associated with a two-to-three-fold increased risk of ovarian carcinoma[42], independently of the known strong association with breast carcinoma.

The remaining proposed HBOC genes with LoF variants present in the cohort (*BLM*, *CHEK2*, *FANCM*, *NBN*, *NF1*, *RAD50*, *RECQL*) have similar or lower frequencies of LoF mutations compared to GnomAD. Whilst these results do not exclude the possibility they may be associated with an increased risk of hereditary ovarian carcinoma, it does suggest that caution should be exercised when interpreting their causative role in the context of germline genetic testing for women with suspected hereditary ovarian carcinoma and no personal or family history of breast carcinoma.

To date, no studies have applied a wholly unbiased WES-based approach to ovarian carcinoma predisposition gene discovery in a case cohort selected for HGSOC and enriched for hereditary cases where *BRCA1* and *BRCA2* involvement have been excluded. Stafford et al.[19] conducted WES on 48 *BRCA1* and *BRCA2*-negative ovarian carcinoma cases with a high prior likelihood of genetic susceptibility, but restricted their candidate gene variant analysis to 155 genes involved in DNA damage response or cell cycle regulation, along with 64 ovarian carcinoma-associated genes listed in the Human Gene Mutation Database (HGMD). Similarly, Lu et al.[20] interrogated WES data from Ambry Genetics for 2051 women with ovarian carcinoma for only a small number of known 'cancer-associated' genes, and demonstrated significant enrichment for variants in six genes (*ATM*, *CHEK2*, *MSH6*, *PALB2*, *RAD51C* and *TP53*). Recently, Zhu et al.[21] analysed WES data from 158 *BRCA1* and *BRCA2*-negative ovarian carcinoma cases and identified *ANKRD11* and *POLE* as putative risk genes following validation studies. Neither gene was found to be enriched for LoF variants in our cohort. However, their analysis of the exome data excluded variants in genes based on expression data and residual variation intolerance scores, and retained predicted pathogenic missense variants. The selective focus of these studies on certain genes also reflects a prevailing assumption about the importance of DNA repair pathway genes in HGSOC that is not supported by our data, which further emphasises the importance of applying an open approach to candidate gene identification.

Other groups alternatively used TCGA germline WES data to search for disease-associated genetic variants, although as noted earlier, this approach has limitations. Kanchi et al.[22], using data from 429 serous ovarian carcinoma TCGA cases and 557 controls, identified several genes enriched for germline deleterious variants that were not previously associated with ovarian carcinoma (e.g. *ASXL1*, *MAP3K1* and *SETD2*). However, their subsequent studies[23,25] did not validate their predisposition gene discoveries. Dicks et al.[24] also used TCGA data from 412 HGSOC cases to identify disease-associated variants in 12 DNA repair genes, and subsequently assessed them in 3107 HGSOC cases and 3368 controls. Of these candidate genes, only *FANCM* had a significantly higher mutation frequency in cases vs. controls. None of the genes identified by Dicks et al. (including *FANCM*) were enriched for LoF mutations in our cohort.

Limitations of the current study include the use of GnomAD as the control population, given the differences in sequencing platforms and variant callers that could result in both false-positive and false-negative associations. Detailed review of variants in the top-ranked genes in both the cases and GnomAD to identify potentially unreliable calls aimed to reduce this problem. While ethnicity differences between the cases and GnomAD exist, these were demonstrated by PCA to be minimal with their predominant (over 95%) Western European ancestry being well matched with the GnomAD NFE non-cancer cohort. In addition, the frequencies of LoF mutations in HBOC genes in GnomAD were broadly comparable to our local population control figures from prior studies[43,44], giving us confidence that in the context of

a gene discovery phase, GnomAD is a suitable surrogate control population.

The largest potential source of uncertainty in this study is the extreme genetic heterogeneity of HGSOC predisposition, with most of the candidate genes only having LoF mutations in less than 0.5% of individuals, meaning that the risk of false-positive associations in the discovery set due to chance or to rare, benign variants will be high (up to 30% for ranked genes with $p$ values < 0.0094 after multiple testing correction). Consequently, it will be essential to conduct further validation using very large case−control studies as well as orthogonal approaches such as tumour sequencing, which can provide powerful evidence of bi-allelic inactivation or other somatic genetic features consistent with the candidate gene actively driving carcinogenesis[45–47].

In summary, WES of the largest cohort of *BRCA1* and *BRCA2*-negative HGSOC cases assembled to date has demonstrated the extensive genetic heterogeneity that exists in the remaining unresolved cases of hereditary HGSOC. Furthermore, the lack of enrichment for LoF mutations in genes either directly or indirectly involved in DNA repair posits an explanation for the lack of success of previous candidate gene studies that have prioritised such classes of genes. This study provides an important, unbiased catalogue of 'function-agnostic' candidate genes based solely on mutation frequencies, which will facilitate additional genetic epidemiological and functional studies with the potential to translate the findings into future clinical practice.

## Methods

**Description of case cohort and controls**. Cases consisted of 516 women from Australia recruited to the Variants in Practice (ViP) study between 2013 and 2018 (Table 5) with a confirmed or suspected diagnosis of HGSOC, as well as those with tumours of similar histology arising in the fallopian tube and peritoneum (which share similar clinical and molecular characteristics to HGSOC and are all thought to originate from foci of serous tubal intraepithelial carcinoma[48]). Represented histological subtypes were high-grade serous (including carcinosarcomas) ($n = 443$); high-grade endometrioid ($n = 35$), which is considered a subtype of HGSOC, distinct from low-grade endometrioid tumours[29,49]; mixed epithelial types with a predominant high-grade serous component ($n = 11$); and suspected high-grade serous tumours that were previously classed as adenocarcinoma not otherwise specified or as unknown ($n = 27$). All women were referred to a specialist familial cancer centre and assessed as fulfilling local criteria for offering of *BRCA1* and *BRCA2* testing (https://www.eviq.org.au/p/620)[50]. Clinical testing for germline variants in both genes was performed using validated, standard techniques (next-generation panel sequencing and/or Sanger sequencing for exon variants, along with multiplex ligation-dependent probe amplification for structural variants) in a certified diagnostic lab, and all tested individuals had no pathogenic or likely pathogenic variants nor any large deletions in these genes. These results were reconfirmed on analysis of their exome sequencing data for *BRCA1* and *BRCA2* pathogenic variants.

Population control frequencies of gene variants were obtained from publicly available sequencing data in GnomAD version 2.1.1 (https://gnomad.broadinstitute.org)[51], containing 125,748 exome sequences and 15,708 genome sequences from unrelated individuals worldwide. Filtering options within GnomAD were used to remove data from individuals with a cancer diagnosis (including those sourced from TCGA) as well as those that were not from a non-Finnish European ethnic background, leaving 59,095 individuals.

**Exome sequencing and variant calling**. Exome sequencing was performed on leucocyte DNA extracted from patient whole-blood samples utilising the Agilent SureSelect (Human All Exon v4 for six samples, and v6 for the remainder) capture and Illumina HiSeq 2500 (150 paired-end reads) sequencing platforms at two commercial sequencing companies (BGI and Novogene). An in-house bioinformatics pipeline constructed using Seqliner v0.7 (http://bioinformatics.petermac.org/seqliner) was used to process raw sequencing data. Raw sequencing reads were quality checked using FastQC v0.11.2 (http://www.bioinformatics.babraham.ac.uk/projects/fastqc), trimmed using cutadapt v1.5 [52] then aligned to the GRCh37/hg19 human reference genome using BWA-MEM v0.7.10 [53]. Duplicate reads were filtered using Picard MarkDuplicates (http://broadinstitute.github.io/picard). Base quality score recalibration and indel realignment were then performed on the filtered reads using the Genome Analysis Toolkit (GATK) v3.8.0 [54]. Variants were called using GATK HaplotypeCaller and Platypus v0.8.1 [55], then annotated for predicted consequences using Ensembl Variant Effect Predictor (VEP) database version v85 [56] and LoFTEE (https://github.com/konradjk/loftee).

**Table 5 Characteristics of total case cohort.**

| | Number (%) |
|---|---|
| Total patients | 516 (100) |
| **Age at diagnosis of ovarian carcinoma** | |
| <30 | 5 (1) |
| 30−39 | 14 (3) |
| 40−49 | 72 (14) |
| 50−59 | 149 (29) |
| 60−69 | 176 (34) |
| 70−79 | 80 (15) |
| ≥80 | 20 (4) |
| **Histopathology** | |
| High-grade serous (incl. carcinosarcoma) | 443 (86) |
| High-grade endometrioid | 35 (7) |
| Mixed epithelial (with predominant high-grade serous component) | 11 (2) |
| Serous (grade unknown/uncertain, presumed high-grade) | 5 (1) |
| Adenocarcinoma NOS (presumed high-grade serous) | 9 (2) |
| Unknown (presumed high-grade serous) | 13 (2) |
| **Personal history of cancer** | |
| Breast (excl. DCIS) | 47 (9) |
| Other (incl. breast DCIS) | 62 (12) |
| No history of cancer | 407 (79) |
| **Family history of ovarian or breast cancer (first- and second-degree relatives only)** | |
| One ovarian cancer case (no breast cancer) | 46 (9) |
| ≥2 ovarian cancer cases (no breast cancer) | 4 (1) |
| One breast cancer case (no ovarian cancer) | 131 (25) |
| ≥2 breast cancer cases (no ovarian cancer) | 47 (9) |
| ≥2 ovarian and breast cancer cases | 34 (7) |
| No known cases of breast or ovarian cancer | 254 (49) |

*NOS* not otherwise specified, *DCIS* ductal carcinoma in situ.

Principal component analysis was performed in PLINK v1.90 [57] using a set of all SNPs passing filters in at least two samples that were targeted by both the Human All Exon v4 and v6 captures and passed linkage disequilibrium pruning ($r^2$ threshold: <0.3, window size: 100 kb, step size: 5 kb). Clusters in the PCA results were classified to ethnicities informed by markers from the major sub-population groups as defined in the GnomAD database.

**Variant filtering, ranking and curation**. A series of filters were applied to the variant data (Fig. 1), using R v3.5.2 (2018) with tidyverse v1.2.1 installed, and the output viewed and analysed in Microsoft Excel v16.25 for Mac. For the discovery analysis, only variants classed by VEP[56] as 'HIGH' impact were retained; these included classic LoF variants (stop-gain, start-loss, frameshift and essential splice site) in protein-coding transcripts, as well as equivalent variants in non-protein-coding transcripts (e.g. non-coding RNAs). Variants classed as 'MODERATE', 'LOW' or 'MODIFIER' impact (including missense, in-frame indel, stop-loss, cryptic splice site, synonymous etc. in protein-coding sequences) were removed. Analysis aimed to identify rare variants with strong pathogenic effect and good-quality sequencing metrics; hence, variants with GnomAD total population minor allele frequency (MAF) > 0.005 or those annotated to non-canonical transcripts (as defined by Ensembl[58,59]) were removed and a number of quality filters applied (Fig. 1). Following ranking (described below), additional filtering removed variants that were not classed as 'protein_coding' in their Ensembl Biotype annotation, leaving only protein-coding LoF variants. Common variants (i.e. MAF > 0.005) in one or more of the major outbred population groups represented in GnomAD (i.e. excluding Finns, Ashkenazi Jewish and 'other' populations) were also removed, using the 'popmax' annotation. The latter filter facilitated the removal of common variants within the other major non-European ethnic groups (e.g. East Asian) represented in the patient sample, abrogating the need to use ethnicity-specific GnomAD data when performing filtering with these cases.

After excluding samples with deleterious variants in known ovarian cancer predisposition genes (Table 1), remaining genes were ranked by degree of enrichment for presumed deleterious variants in the case population. To facilitate this, total control population frequencies of 'HIGH' impact variants for every gene transcript were calculated using the GnomAD non-cancer reference data for the non-Finnish European (NFE) sub-population[51]; these figures excluded common variants with MAF > 0.005, and were adjusted for genes with multiple variants per individual using the formula $1 - \prod(1 - AF_i)$ i.e. one minus the combined probability of not containing any of the variant alleles. Variants that were flagged in

GnomAD as failing their 'InbreedingCoeff', 'AC0' or 'RF' (random forests) QC filters were excluded from these figures, to match our filtering. Total frequencies for every gene with retained variants in the sample were calculated, and a risk ratio between figures for the two population groups (case cohort and GnomAD non-cancer NFE) was derived.

A two-tailed chi-squared test was then used to compare the total number of rare (i.e. $AF \leq 0.005$) LoF variants in the case cohort vs. the equivalent number in the GnomAD non-cancer NFE sub-population for all genes represented on the Agilent SureSelect v6 exome panel with 'protein_coding' Biotype transcripts ($n = 19,818$). $p$ values, odds ratios (ORs) and confidence intervals for every gene were then calculated using a two-tailed Fisher's exact test, incorporating allele counts in the sample vs. equivalent counts in the GnomAD non-cancer NFE sub-population (with the denominator as the maximum number of alleles from that population with available data in GnomAD for that specific gene). Genes were ranked in order of increasing $p$ value, with the most enriched genes having the lowest $p$ values, and the calculated risk ratios were used to prioritise variants in genes that were enriched by three-fold or more in the case cohort for further analysis. Additional two-tailed chi-squared tests were used to compare the observed vs. the expected distribution of Fisher's test $p$ values < 0.01 for odds ratios >1 and <1 for genes with 'protein_coding' Biotype transcripts. The Benjamini−Hochberg procedure[28] for multiple testing was applied to the ranked list of $p$ values to establish a 'discovery' threshold $p$ value for prioritising top-ranked genes for further study, specifying a false discovery rate of 0.3. It is important to note that the $p$ values used for ranking candidate genes do not imply a statistically significant difference in total LoF allele frequency between cases and the GnomAD sub-population for any individual gene, since the case cohort lacked the size and power required to demonstrate this. A two-tailed Fisher's exact test was also used to compare the total frequency of LoF variants in known DNA repair genes grouped by functional pathway (from Chae et al.[30]) in the discovery cohort ($n = 510$) with those in the GnomAD non-cancer NFE sub-population; this analysis did not include $BRCA1$ and $BRCA2$ or any of the other known ovarian carcinoma predisposition genes that had been analysed for LoF variants in the case cohort during filtering (described below). All graphs were plotted using GraphPad Prism v8.1.1 for Mac, and all statistical tests (Fisher's exact test, chi-squared tests and the Benjamini−Hochberg procedure) were performed in R or Prism.

Ranked genes and LoF variants were curated and scrutinised using available online databases (NCBI Gene, OMIM and COSMIC[38]) to annotate their function and possible role in cancer predisposition. GnomAD data for each gene were also reviewed, to identify those genes with problematic sequencing data, or variants that were found at an $AF > 0.005$ in one of the GnomAD sub-populations; any genes or variants affected as such were excluded from the top-ranked gene list. Finally, the candidate gene variants with borderline quality sequencing metrics (i.e. failed QC sequencing quality score < 500, read depth < 60, alt allele read frequency < 0.35 or variants not called bidirectionally) were manually reviewed within the raw sequencing (bam) files using the Integrative Genomics Viewer (IGV) software[60]; any doubtful variants were excluded when collating the top-ranked gene list. A two-tailed Fisher's exact test was used at this point to compare the likelihood of being a candidate gene carrier in those with a family history of breast and/or ovarian cancer in one or more first- or second-degree relatives ($n = 262$) vs. those with no family history ($n = 254$).

**Analysis of known and proposed ovarian carcinoma risk genes**. For known and proposed ovarian carcinoma predisposition genes ($MLH1$, $MSH2$, $MSH6$, $PMS2^5$, $BRIP1^9$, $RAD51C^7$, $RAD51D^8$, $PALB2^{61}$, $FANCM^{24}$, $ATM^{14}$, $TP53^{62}$, $CHEK2^{63}$, $BARD1^{64}$, $STK11^{65}$, $CDH1^{66}$, $PTEN^{67}$, $FANCC^{68}$, $RECQL^{69}$, $BLM^{68}$, $NF1^{70}$ and the MRN protein complex genes i.e. $MRE11A$, $NBN$, $RAD50^{71}$), any identified LoF variants annotated to RefSeqGene transcripts[72] were considered pathogenic and retained, but additionally checked in NCBI ClinVar (https://www.ncbi.nlm.nih.gov/clinvar) to exclude any that had been classed in this database as 'benign' or 'likely benign'. Only missense variants classed as pathogenic in ClinVar with multiple sources of supporting evidence and consensus opinion were considered deleterious.

**Ethics statement**. This study protocol was approved by the Human Research Ethics Committees at each participating ViP study recruitment centre and the Peter MacCallum Cancer Centre (approval nos. 11/50 and 09/29). All participants provided informed consent for genetic analysis of their germline and tumour DNA.

**Reporting summary**. Further information on research design is available in the Nature Research Reporting Summary linked to this article.

## Data availability
The exome sequencing data have been deposited in the European Genome-phenome Archive under the study ID EGAS00001004235 and the dataset accession code EGAD00001006030, and is available upon request on application to the linked Data Access Committee at dac@petermac.org (https://www.ebi.ac.uk/ega/dacs/EGAC00001001505). Other datasets referenced during the study are available from the GnomAD (https://gnomad.broadinstitute.org/) and COSMIC (https://cancer.sanger.ac.uk/cosmic) websites. All other data supporting the findings of this study are available within the article and its Supplementary Information files and from the corresponding author upon reasonable request. A reporting summary for this article is available as a Supplementary Information file.

## Code availability
R script used for data analysis available at https://rpubs.com/deepsubs/nature_comms_paper_2020. The Seqliner code is available separately from the R script at http://bioinformatics.petermac.org/seqliner/. All other publicly available code used during exome sequence data processing and variant calling are available via the links mentioned within the methods.

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

## Acknowledgements

The authors wish to thank the staff at the Victorian and Tasmanian Familial Cancer Centres who enrolled participants and provided clinical data, and we thank all the participants of the ViP study for donating their DNA samples and consenting to share their clinical information. D.N.S. wishes to thank the staff in the Bioinformatics Core Facility as well as Dr. Elizabeth Christie and Prof. Ingrid Winship for their advice and support. This work was supported by a National Health and Medical Research Council Program Grant (APP1092856 to I.G.C. and P.A.J.) and Medical/Dental Postgraduate Scholarship (GNT1134107 to D.N.S.), and an Australian Government Research Training Program Scholarship.

## Author contributions

D.N.S. was responsible for formal data analysis, investigation, validation and writing of the original draft manuscript, and contributed to data curation, methodology and visualisation. M.Z. and N.L. contributed to methodology and software, with M.Z. additionally contributing to data curation, visualisation and writing of the draft manuscript. S.M., J.A.M., S.M.R. and J. E.A.L. all contributed to data curation; S.M. and S.M.R. were additionally responsible for project administration, and J.A.M. and J.E.A.L. separately contributed to software and investigation, respectively. I.G.C. and P.A.J. were jointly responsible for project conceptualisation, funding acquisition, supervision and methodology, with I.G.C. providing the resources for this work. I.G.C., P.A.J. and K.L.G. reviewed and edited the final manuscript.

## Competing interests

The authors declare no competing interests.
