## [Peer Review File · Nature Communications]

Reviewers' comments:

Reviewer #2 (Remarks to the Author): Expertise in ovarian cancer risk

Subramamian et al aimed to discover new genes for ovarian cancer by conducting an exome sequencing study in a large number of women seen in the family cancer clinic in Melbourne and tested for BRCA1 and BRCA2, with negative results.

They conclude (rather like the Chubb et al paper on CRC, published by Nat Commun some time ago) that there are no more major genes to discover in HGSC, and what is left of the heritable component of ovarian cancer is both very rare and, in general, not attributable to DNA repair genes. This is a helpful contribution to the literature. There are no more genes like RAD51C/D and BRIP1 to be found. The paper also supports an etiological role for ATM and PALB2.

Some recommendations for strengthening the manuscript

1) study of the paired tumors in cases where for example the OR is greater than 5, and more than 5 LOF variants are found, would definitely add a lot to this ms, which is otherwise essentially negative, in terms of identifying strong candidate genes. This is mentioned on page 12, but is not offered in this paper

2) discussion of result of segregation studies in the variants of interest in familial cases should be undertaken

3) the MRE11A results worry me. There have been numerous studies to suggest that MRE11A, and its partners NBN/RAD50 in the MRN complex, are ovarian (or breast) cancer susceptibility alleles. Nearly all of the studies have been null, but now this study puts MRE11A back into play. I strongly urge the authors to work with well known MRE11A experts such as John H. J. Petrini, who could add significantly to this paper, before publishing these data.

Minor points

The title is a bit confusing to me. Is this a study of High grade serous carcinoma of the ovary, or is this a study of Epithelial ovarian cancer, and then HGSC tumors were favored for analysis? The title is about HGSC (the correct final term is carcinoma, not cancer, by the way) but the abstract talks about "EOC" which is a ragbag of different disorders. It would be helpful for a pathologist to be more obviously involved in this study.

Ref 4 talks about 19% of women with "ovarian cancer" having germline pathogenic BRCA1/2 variants, but again I am not clear what this means exactly. I can't find that number directly in the paper. HGSC? EOC? other?

The intro also says MMR genes contribute to EOC, but they don't contribute to HGSC, so again the definitions are imprecise here.

"None of the previous studies have identified any compelling novel candidate genes" - what is meant by "compelling" or "novel" - does FANCM count, for example? If not, why not?

Reviewer #3 (Remarks to the Author): Expertise in cancer genomics

The manuscript by Subramanian, Campbell and colleagues reports their findings from exome sequencing of a cohort of 516 BRCA1/2 negative women with ovarian cancer (largely high grade serous) and a family history of disease. The cohort represents one of the largest to be approached with an 'unbiased' exome approach. The findings, although unvalidated, suggest that there could be substantial predisposition in genes/pathways not directly associated with DNA repair. This would expand the potential biological landscape of risk in ovarian cancer, and would further instantiate the genetic heterogeneity of the disease.

There are several points and critiques that are important to highlight as opportunities to improve the study.

- 1) The findings are not validated in an independent cohort, and thus serve to highlight a potentially interesting set of findings rather than more definitive results. While it is, of course, quite easy to point this out as a deficiency it is important given the departure from the more deeply explored DNA repair space and its implications.
- 2) Re BRCA1/2 negativity, was this based solely on a clinical exon-targeted assessment, or was there a survey of larger rearrangements/structural mutations included in the ruling out?
- 3) It would be good to have more detail into what constitutes 'familial' in this cohort? Was there any breakdown of BRCA1/2 positivity versus not in the extent of family history? Was there any skew in family history between the ultimate potential novel predisposition gene and the rest of the cohort?
- 4) Is there any association of rare suspect gene and gene size? More broadly, it might of some use to conduct permutation-based testing to assess (observed versus expected) this many deleterious variants would be expected to be seen in a random sample from the Gnomad or TCGA type data sets. The authors are to be commended for addressing the not insubstantial likelihood of false positives even amongst the ranked genes. This may be at least one approach to further elaborate on this aspect.
- 5) Although population stratification is addressed to an extent by the PCA analyses (with the margin of positives still somewhat within the error), is there another approach to ask whether the ranked genes are more or less likely to fall on rare(r) haplotypes using imputation approaches? The implications of the common control group and likelihood of false positive re frequencies are a significant concern (one that the authors do acknowledge).
- 6) Is the interpretation that (from page 7) that in the final set of 66 top recurrent/multiple testing corrected candidate genes, fully a third were found to be low quality upon manual curation correct? With the caveat that recurrence can be a surrogate for sequencing artifact, it would be good for the investigators to comment on the implications for the veracity of the calling pipeline being used and the potential number for problematic calls manifesting at all levels (and thus in the outputs).
- 7) For the genes implicated (particularly the LOF), is there any evidence that there is significant association with somatic LOH in these genic regions in "sporadic" or other data available? Are there somatic mutations reported in these genes in ovarian cancer datasets, and if so, is there any evidence for enrichment (eg are these significantly mutated genes when you remove TP53 and few other recurrent gene from the statistic?) or for these genes being somatically mutated? If not, could the authors speculate on why this might be?
- 8) Have the authors attempted a more standardized rare allele variant burden test? If so, what are there results? If not, this may be another approach given the case/control mismatch in size and matching per se.

Responses to Reviewers' Comments

Reviewer #2

1. *'study of the paired tumors in cases where for example the OR is greater than 5, and more than 5 LOF variants are found, would definitely add a lot to this ms, which is otherwise essentially negative, in terms of identifying strong candidate genes. This is mentioned on page 12, but is not offered in this paper'*

Although ultimately tumour sequencing may prove helpful, sourcing sufficient numbers of archival cancer samples is an enormous task (and for many patients is impossible because their tumour blocks have been discarded over time), and is well beyond the scope of the paper. Furthermore, all of the candidate genes with $OR > 5$ have LOF variants present in only a small number of individuals (the vast majority in only three to four), and in our experience even fewer have accessible blocks (i.e. one to two per gene). In order to generate significant data for validating our top candidate genes, far more individuals and blocks would be required, which might only be possible as part of a future international collaborative effort.

2. *'discussion of result of segregation studies in the variants of interest in familial cases should be undertaken'*

Again, we feel that this type of work is beyond the scope of this paper. As well as similar issues relating to time constraints and logistical challenges, relatively few families have sufficiently informative pedigrees for conducting meaningful segregation. This problem is further compounded by the difficulty of recruiting sufficient numbers of affected relatives due to the poor survival rate of HGSOc.

3. *'the MRE11A results worry me. There have been numerous studies to suggest that MRE11A, and its partners NBN/RAD50 in the MRN complex, are ovarian (or breast) cancer susceptibility alleles. Nearly all of the studies have been null, but now this study puts MRE11A back into play. I strongly urge the authors to work with well-known MRE11A experts such as John H. J. Petrini, who could add significantly to this paper, before publishing these data'*

The *MRE11A* results, although marginally 'significant', were near the bottom of the ranked list and not in the top 66 that passed multiple testing correction, with only three carriers detected. To imply that this brings it 'back into play' without further supporting evidence (e.g. from tumour sequencing studies) is an over-interpretation of data that might be a false-positive result. We do not consider three carriers as sufficient to justify functional studies or the involvement of a *MRE11A* expert. Our paper makes this clear in the Results section (p6), where we state that *'...the number of variants and cases was small, and caution should be exercised interpreting the odds ratios as risk estimates.'* We agree with the reviewer that the evidence for *MRE11A* as a HBOC gene is weak, with two recent large case-control studies (Norquist *et al*, 2016; Lilyquist *et al*, 2017) showing no association with ovarian carcinoma in particular.

4. *'The title is a bit confusing to me. Is this a study of High grade serous carcinoma of the ovary, or is this a study of Epithelial ovarian cancer, and then HGSC tumors were favored for analysis? The title is about HGSC (the correct final term is carcinoma, not cancer, by the way) but the abstract talks about "EOC" which is a ragbag of different disorders. It would be helpful for a pathologist to be more obviously involved in this study.'*

This was a study of individuals with confirmed or suspected high-grade serous ovarian tumours, or high-grade serous-like ovarian tumours (as described in the Methods, p15). We have gone through the Abstract and main text and made several changes to make it clearer that this was the tumour type analysed. All references to 'cancer' have also been changed to 'carcinoma', where appropriate. We feel these textual changes are sufficient to negate any requirement to include a pathologist in the authorship.

5. *'Ref 4 talks about 19% of women with "ovarian cancer" having germline pathogenic BRCA1/2 variants, but again I am not clear what this means exactly. I can't find that number directly in the paper. HGSC? EOC? other?'*

The quoted figure and references have been amended in the relevant paragraph on p3 as follows (**bold text**):

*'There is a significant genetic component to the risk of ovarian carcinoma³, **with germline mutations in BRCA1 and BRCA2 identifiable in 11-23% of affected women with HGSOC^{4,5}**, rising to as high as 42% of affected women with a family history of two or more ovarian carcinomas⁶.'*

6. *'The intro also says MMR genes contribute to EOC, but they don't contribute to HGSC, so again the definitions are imprecise here.'*

We have clarified the contribution of MMR genes to ovarian carcinoma in the text (p3 and 10), making it clearer that these genes are predominantly associated with non-HGS ovarian carcinoma tumour types.

7. *'"None of the previous studies have identified any compelling novel candidate genes" - what is meant by "compelling" or "novel" - does FANCM count, for example? If not, why not?'*

'Novel' gene = one not previously proposed as a germline predisposition gene for that particular tumour type (in this instance, HGSOC). 'Compelling' gene = one supported by multiple independent studies using different lines of evidence (i.e. epidemiological and functional studies) and thus widely accepted as a cancer predisposition gene by experts in the field. We agree the latter is a very subjective term, and have opted to remove it from the relevant sentence on p4, which now reads as follows:

'None of the previous studies have identified any novel candidate HBOC genes that have been validated in multiple independent studies; nor has there been any consistency of the candidates identified across different studies.'

FANCM, whilst 'novel', is only supported by one study (Dicks *et al*, 2017) in HGSOE with no further supporting evidence from other unrelated studies. We have kept the term 'novel' in the sentence as we believe this is the most appropriate description and does not justify replacing it with an extended text description.

Reviewer #3

1. 'Re *BRCA1/2* negativity, was this based solely on a clinical exon-targeted assessment, or was there a survey of larger rearrangements/structural mutations included in the ruling out?'

Testing for *BRCA1/2* pathogenic mutations was performed using a combination of next generation panel sequencing and/or Sanger sequencing for exon-targeted assessment, in combination with multiplex ligation-dependent probe amplification for larger rearrangements and other structural mutations. Furthermore, all included individuals had no detectable pathogenic LOF variants in these genes within their exome sequencing data. These details have been added to the Methods section (p15):

'Clinical testing for germline variants in both genes was performed using validated, standard techniques (next generation panel sequencing and/or Sanger sequencing for exon variants, along with multiplex ligation-dependent probe amplification for structural variants) in a certified diagnostic lab, and all tested individuals had no pathogenic or likely pathogenic variants nor any large deletions in these genes. These results were reconfirmed on analysis of their exome sequencing data for *BRCA1* and *BRCA2* pathogenic variants.'

2. 'It would be good to have more detail into what constitutes 'familial' in this cohort? Was there any breakdown of *BRCA1/2* positivity versus not in the extent of family history? Was there any skew in family history between the ultimate potential novel predisposition gene and the rest of the cohort?'

'Familial' in this context refers to individuals with a family history of ovarian and/or breast cancer in at least one first or second degree relative, which comprised over 50% of our total cohort (see Table 1). Furthermore, our method of case ascertainment and recruitment, performed through a familial cancer centre clinic and restricted to those patients who met local criteria for *BRCA1/2* testing (<https://www.eviq.org.au/p/620>) maximised the likelihood of including other individuals with a hereditary predisposition to cancer who do not have a family history of ovarian and/or breast cancer due to their pedigree structure (i.e. small, male-predominant family), or a *de novo* mutation. Taken together, we consider this cohort to be substantially 'enriched' for familial cases, which is why we have described it as such in the title as well as throughout the paper. We have added the above link to the eviQ guidelines for *BRCA1/2* testing in the Methods (p 15) as follows:

'All subjects were referred to a specialist familial cancer centre and assessed as fulfilling local criteria for offering them *BRCA1* and *BRCA2* testing (<https://www.eviq.org.au/p/620>)⁴⁹.'

No data is available for *BRCA1/2* positivity in the context of family history, since only *BRCA1/2*-negative individuals were included in the study. We did analyse the family history distribution (divided into 'no family history' and 'family history of at least one breast or ovarian cancer in a first or second-degree relative') of candidate gene carriers (n = 138) versus non-carriers (n = 378), and found no significant difference using Fisher's exact test ($p = 0.55$). This has been added to the Results (p7) as follows:

'Comparing the family history distribution of candidate gene carriers (n =138) and non-carriers (n = 378) using Fisher's exact test, there was no significant difference in the likelihood of being a carrier in those with a family history of breast and/or ovarian cancer in at least one first or second degree relative versus those with no family history ($p = 0.55$).'

3. *'Is there any association of rare suspect gene and gene size? More broadly, it might of some use to conduct permutation-based testing to assess (observed versus expected) this many deleterious variants would be expected to be seen in a random sample from the Gnomad or TCGA type data sets. The authors are to be commended for addressing the not insubstantial likelihood of false positives even amongst the ranked genes. This may be at least one approach to further elaborate on this aspect.'*

Since we are comparing the frequency of variants in the candidate genes with the frequency in the same gene reported in Gnomad, we feel that in the context of our analysis, the candidate gene size is irrelevant (i.e. for large genes, the GnomAD LOF mutation frequency will also be higher). Similarly, permutation-based testing as described above is more appropriate for case-only studies, and would not provide any additional insight into our results beyond that provided by our included comparison with the GnomAD data set.

4. *'Although population stratification is addressed to an extent by the PCA analyses (with the margin of positives still somewhat within the error), is there another approach to ask whether the ranked genes are more or less likely to fall on rare(r) haplotypes using imputation approaches? The implications of the common control group and likelihood of false positive re frequencies are a significant concern (one that the authors do acknowledge).'*

To address the reviewer's comments, we have replotted the PCA plot (**Supplementary Figure 1**) to highlight those cases with candidate gene variants (filled symbols) versus those without (empty symbols). This shows that the cases harbouring candidate gene LOF variants were not outliers in relation to their imputed ancestry and to the PCA plot as a whole. We have also added additional information to the Supplementary Methods (p1) describing how ethnicity in our cohort was assigned based on the PCA results, as follows:

'Clusters in the PCA results were classified to ethnicities informed by markers from the major subpopulation groups as defined in the GnomAD database.'

PCA is the best method for assessing ancestry in this type of study, and haplotype analysis as described above would only be useful for examining multiple occurrence of the same variant in our data (i.e. by comparing adjacent alleles in an ancestral haplotype block in our cases versus GnomAD). As the GnomAD database does not contain haplotype data for

individual exomes/genomes or variants, we are unable to perform this analysis using our chosen control data set. Besides, the vast majority (80%) of individual candidate gene variants occurred only once in our cohort.

5. *'Is the interpretation that (from page 7) that in the final set of 66 top recurrent/multiple testing corrected candidate genes, fully a third were found to be low quality upon manual curation correct? With the caveat that recurrence can be a surrogate for sequencing artifact, it would be good for the investigators to comment on the implications for the veracity of the calling pipeline being used and the potential number for problematic calls manifesting at all levels (and thus in the outputs).'*

All variants were called using PLATYPUS and HAP, the latter of which forms part of the gold-standard GATK pipeline used by multiple other research groups worldwide. Although it is correct to infer that a third of genes with problematic data were removed following manual review, this is not the same as saying that a third of all called variants within the top-ranked candidate genes were 'low-quality'. Only 59 variants out of the 283 (21%) in the top-ranked 66 genes were judged to be 'doubtful', due to issues identified in our own sequencing data or the equivalent data in GnomAD i.e. we only manually curated top-ranked variants that had marginal quality or other metrics (as described in the Supplementary Methods, p4-5), hence the higher fall out rate for this small subset out of the total number of called variants (over 10,000) that remained following the initial and further filtering processes summarised in Figure 1. Thirty-one additional variants (11%) were excluded not because of quality issues, but because of their high frequency in one of the GnomAD major sub-populations (i.e. AF > 0.005). Finally, after these problematic variants were removed, fifteen of the remaining genes had 22 valid variants but no longer met our three-sample/three-fold risk ratio thresholds, and so were excluded from the final candidate list on this basis alone. These points have been clarified in the Results (p7) and Supplementary Methods (p4-5) as follows:

Results, p7:

'Twenty-three genes with low-confidence LoF variants were removed during curation; these included fifteen genes that were removed due to their remaining valid variants occurring in fewer than three individuals, or falling below our three-fold enrichment threshold.'

Supplementary Methods, p4-5:

'GnomAD data for each gene was also reviewed, to identify those genes with problematic sequencing data, or variants that were found at an AF above 0.005 in one of the GnomAD sub-populations; any genes or variants affected as such were excluded from the top-ranked gene list.'

To our knowledge, previous studies have not curated their gene lists to this level of detail prior to publishing candidate gene variants, and we regard this as a major strength of the paper. In addition, since our multiple correction false positive threshold of 0.3 was applied *prior* to our additional filtering and manual curation steps, we feel confident that we have minimised the number of remaining 'false-positive' results in the candidate gene list through these measures.

6. *'For the genes implicated (particularly the LOF), is there any evidence that there is significant association with somatic LOH in these genic regions in "sporadic" or other data available? Are there somatic mutations reported in these genes in ovarian cancer datasets, and if so, is there any evidence for enrichment (eg are these significantly mutated genes when you remove TP53 and few other recurrent gene from the statistic?) or for these genes being somatically mutated? If not, could the authors speculate on why this might be?'*

To answer this query, we have searched the COSMIC database (which includes the TCGA dataset, amongst others) for pathogenic somatic candidate gene mutations in serous ovarian tumour samples. The findings have been added to Table 4 and commented on in the Results (p7) and Discussion (p10) as follows:

Results, p7:

'The majority of these candidate genes have not been reported to contain pathogenic somatic mutations in serous ovarian tumour samples from the COSMIC database (Table 4), and for those that do, the frequency of somatic variants is low (<1% of samples).'

Discussion, p10:

'Only sixteen top-ranked genes had any somatic mutations recorded in COSMIC (Table 4), none of which exceeded 1% of serous ovarian tumour samples present in the database. This is consistent with the finding that established germline susceptibility genes, with the exception of TP53, are also rarely found to harbour somatically-acquired mutations in sequenced tumour samples. BRCA1 pathogenic somatic mutations, for example, are only present in 1.59% of serous ovarian carcinoma samples in the COSMIC database³⁸.

To summarise, mutations within these individual predisposition genes in a sporadic tumour precursor cell might be considered as very rare events, in contrast to the essential loss of function of certain 'gatekeeper' genes such as TP53. Hence, whilst the presence of somatic pathogenic mutations within a candidate gene is supportive of an association, their absence does not necessarily exclude it from consideration as a germline contributor. Consequently, testing for enrichment for somatic LOF variants within these candidate genes in the manner the reviewer proposes is unlikely to reveal any significant findings.

Whilst it would certainly be interesting to see if there was evidence of somatic LOH for these candidate genes in sporadic tumour samples, the absence of associated germline sequencing data for most of the publicly available datasets in COSMIC precludes us from performing this analysis.

7. *'Have the authors attempted a more standardized rare allele variant burden test? If so, what are their results? If not, this may be another approach given the case/control mismatch in size and matching per se.'*

We have not attempted a rare allele variant burden test, since the tools we are aware of for performing this (e.g. SKAT, WST) require a matrix of haplotypes as input, which we are unable to provide in the absence of any haplotype or individual sample information from the GnomAD control data set.

REVIEWERS' COMMENTS:

Reviewer #2 (Remarks to the Author):

I do not have much to add to my original comments as the authors have opted not to perform any more experiments.

I have a few remaining points

1) the title - on reading over the paper again, I wondered if it might be more informative to title the paper

"Exome sequencing of familial high-grades serous ovarian carcinoma reveals extreme genetic heterogeneity for rare candidate susceptibility genes"

as the current title does not give much of an inkling of the results.

2) what does "they" refer to in line 215? Specifically, are the authors referring to PALB2, ATM, and MRE11A, or other variants found after these were excluded.

3) Please consider discussing PALB2 as an ovarian cancer gene - see JCO paper from Tischkowitz that just came out confirming PALB2 as an ovarian cancer gene. The evidence for ATM is compelling but less certain. So I think PALB2, ATM and MRE11A all represent different classes of mutations. Please cite the papers mentioned in the response letter that show MRE11A is not yet associated with increased risk for ovarian cancer.

Reviewer #3 provided no further comments for the authors to address.

Responses to Reviewers' Comments

Reviewer #2

- 1) *'the title - on reading over the paper again, I wondered if it might be more informative to title the paper*

"Exome sequencing of familial high-grades serous ovarian carcinoma reveals extreme genetic heterogeneity for rare candidate susceptibility genes"

as the current title does not give much of an inkling of the results.'

We have amended the title per the editor's suggestion as follows:

"Exome sequencing of familial high-grade serous ovarian carcinoma reveals heterogeneity for rare candidate susceptibility genes"

- 2) *'what does "they" refer to in line 215? Specifically, are the authors referring to PALB2, ATM, and MRE11A, or other variants found after these were excluded.'*
- 3) *'Please consider discussing PALB2 as an ovarian cancer gene - see JCO paper from Tischkowitz that just came out confirming PALB2 as an ovarian cancer gene. The evidence for ATM is compelling but less certain. So I think PALB2, ATM and MRE11A all represent different classes of mutations. Please cite the papers mentioned in the response letter that show MRE11A is not yet associated with increased risk for ovarian cancer.'*

"They" refers to genes *other* than those mentioned above; specifically, *BLM, CHEK2, FANCM, NBN, NF1, RAD50* and *RECQL*. To address this as well as the reviewer's other comments, we have rewritten the relevant paragraph (p10 – 11) as follows, with additions and changes highlighted in bold:

"Although many of the suspected HBOC-associated genes harboured LoF variants (Table 3), the frequencies were low and only *PALB2, ATM* and *MRE11A* showed some degree of enrichment compared to GnomAD. **The level of enrichment was relatively modest, with very wide confidence intervals due to the small numbers present, making it challenging to interpret their true significance. Previous work suggested a similarly modest increase in risk for *ATM* and *PALB2*, but not for *MRE11A*^{13,14}, casting doubt on whether the latter is truly an ovarian carcinoma predisposition gene. Recent additional data from *PALB2* families found that pathogenic variants are associated with a two-to-three-fold increased risk of ovarian carcinoma⁴², independently of the known strong association with breast carcinoma.**

"The remaining proposed HBOC genes with LoF variants present in the cohort (*BLM, CHEK2, FANCM, NBN, NF1, RAD50, RECQL*) have similar or lower frequencies of LoF mutations compared to GnomAD. Whilst these results do not exclude the possibility they may be associated with an increased risk of hereditary ovarian carcinoma, it does

suggest that caution should be exercised when interpreting their **causative** role in the context of germline genetic testing for women with suspected hereditary ovarian carcinoma and no personal or family history of breast **carcinoma.**”